# Description and analysis of representative COVID-19 cases–A retrospective cohort study

Yannis Herrmann[1]☯*, Tim Starck[2]☯, Niall Brindl🄳[1], Philip J. Kitchen🄳[1], Lukas Rädeker[1], Jakob Sebastian[1], Lisa Köppel🄳[3], Frank Tobian[3], Aurélia Souares[2], André L. Mihaljevic[4], Uta Merle[5], Theresa Hippchen[5], Felix Herth[6], Britta Knorr[7], Andreas Welker[7], Claudia M. Denkinger🄳[2]*

**1** Faculty of Medicine, Ruprecht Karl University of Heidelberg, Heidelberg, Germany, **2** Heidelberg Institute of Global Health, Heidelberg University Hospital, Heidelberg, Germany, **3** Division of Tropical Medicine, Centre of Infectious Diseases, Heidelberg University Hospital, Heidelberg, Germany, **4** Department of General, Visceral and Transplantation Surgery, Heidelberg University Hospital, Heidelberg, Germany, **5** Department of Gastroenterology, Heidelberg University Hospital, Heidelberg, Germany, **6** Department of Pneumology and Critical Care Medicine, Chest Hospital, Heidelberg University Hospital, Heidelberg, Germany, **7** Department of Public Health, Landratsamt Rhein-Neckar, Heidelberg, Germany

☯ These authors contributed equally to this work.
* Claudia.denkinger@uni-heidelberg.de (CMD); y.herrmann@stud.uni-heidelberg.de (YH)

**Data Availability Statement:** All relevant data are within the manuscript and its Supporting Information files.

## Abstract

### Background

Most data on COVID-19 was collected in hospitalized cases. Much less is known about the spectrum of disease in entire populations. In this study, we examine a representative cohort of primarily symptomatic cases in an administrative district in Southern Germany.

### Methods

We contacted all confirmed SARS-CoV-2 cases in the administrative district. Consenting participants answered a retrospective survey either via a telephone, electronically or via mail. Clinical and sociodemographic features were compared between hospitalized and non-hospitalized patients. Additionally, we assessed potential risk factors for hospitalization and time to hospitalization in a series of regression models.

### Results

We included 897 participants in our study, 69% out of 1,305 total cases in the district with a mean age of 47 years (range 2–97), 51% of which were female and 47% had a pre-existing illness. The percentage of asymptomatic, mild, moderate (leading to hospital admission) and critical illness (requiring mechanical ventilation) was 54 patients (6%), 713 (79%), 97 (11%) and 16 (2%), respectively. Seventeen patients (2%) died. The most prevalent symptoms were fatigue (65%), cough (62%) and dysgeusia (60%). The risk factors for hospitalization included older age (OR 1.05 per year increase; 95% CI 1.04–1.07) preexisting lung conditions (OR 3.09; 95% CI 1.62–5.88). Female sex was a protective factor (OR 0.51; 95% CI 0.33–0.77).

**Funding:** This work was supported by internal COVID funds of the Heidelberg University Hospital.

**Competing interests:** The authors have declared that no competing interests exist.

## Conclusion

This representative analysis of primarily symptomatic COVID-19 cases confirms age, male sex and preexisting lung conditions but not cardiovascular disease as risk factors for severe illness. Almost 80% of infection take a mild course, whereas 13% of patients suffer moderate to severe illness.

## Trial registration

German Clinical Trials Register, DRKS00022926. URL: https://www.drks.de/drks_web/setLocale_EN.do

## Introduction

SARS-CoV-2 has affected the entire globe with millions of confirmed cases, leading to increasing fatalities [1]. Germany is among the most affected countries worldwide [2].

Previous studies focused mostly on the clinical features and potential risk factors for a severe course of COVID-19 in hospitalized cases. The majority of infected patients, however, remain asymptomatic or suffer mild symptoms and recover in home-quarantine. This group of patients is underrepresented in most studies. Furthermore, few countries other than Germany offered the extensive testing capacity to identify almost all symptomatic infections. Thus a high number of unreported cases has to be presumed based on the observed COVID-19 specific and excess mortality in studies from countries like Italy, the United Kingdom or United States of America [3, 4]. The paucity of representative cohorts makes it difficult to draw conclusions in regard to the typical sociodemographic and clinical features of COVID-19 as well as risk factors for severe disease.

The few available studies examining whole cohorts were linked to distinct settings with selected patient populations (e.g. homeless shelter, cruise ship) [5–7]. The proportion of persons infected developing severe disease in these and other large cohort studies was estimated to be around 16–24% [6, 8–10]. Identified risk factors for severe disease in hospitalized patients include advanced age, male sex and comorbidities such as hypertension, cardiovascular disease, diabetes or COPD [11–15].

In our study presented here, we describe clinical features, demographics, epidemiological characteristics and assess potential risk factors of severe disease for a cohort of persons infected with SARS-CoV2 in an entire administrative district in Southern Germany. Due to the widespread case finding and testing capabilities in Germany and the region especially, our study represents a near-complete population cohort reflecting the entire spectrum of disease [2].

## Methods

### Study design and participants

The study took place in the Rhein-Neckar region of Germany from March 19, 2020 until June 30, 2020. The Rhein-Neckar-Region inhabits approximately 710,000 people, constituting an administrative district of Germany [16, 17]. One of the largest University hospitals in Germany, the Heidelberg University Hospital is located in this district with an approximate of 2000 beds including 156 beds with mechanical ventilation. At no point during the pandemic were the capacities of the University Hospital and surrounding hospitals exceeded. A national lock-down in Germany was announced on March 15, 2020 and lasted until April 19th.

Persons of all age groups who tested positive for SARS-CoV-2 using RT-PCR nasopharyngeal swabs, identified through the registry of the public health authority in the Rhein-Neckar-Region in Germany between February 7 and June 30, 2020 were screened and asked for consent. At the time of the study, testing was performed based on clinical suspicion, i.e. presence of symptoms or high risk contact. If a participant was not able to give written consent due to death or legal care, we asked first degree relatives or guardians to fill out the survey on behalf of the participant. Consenting participants were contacted after they had completed two weeks of quarantine and were asked to fill out a survey developed by infectious disease clinicians based on findings in the literature (S1 File). Overall data on number of cases and number of cases hospitalized and COVID-related deaths were available from the public health authority. Data protection was in line with the German data protection laws and the General Data Protection Regulation of the European Union.

## Data collection

After confirming consent, participants were invited to participate via a phone interview or an electronical or paper-based questionnaire was sent to them to fill out. All data used in this study was collected with the retrospective survey, no other sources were used.

The Research Electronic Data Capture (REDCap^TM, www.prorect-redcap.org) hosted at the University Hospital Heidelberg was used for data management. REDCap^TM is a secure, web-based application, which provides audit trails for tracking data manipulation and export procedures [18].

## Data, variables and definitions

**Variables.**   We collected sociodemographic, clinical variables and outcome indicators. The questionnaire and translation of assessed information is available in the supplement (S1 File).

**Outcome indicators.**   For the descriptive analysis we stratified our study population by a five-level categorical outcomes variable: (1) Asymptomatic cases were defined as patients with confirmed SARS-CoV-2 infection who did not report any symptoms or clinical signs over the course of their duration of quarantine. (2) Symptomatic outpatients requiring no hospital admission were considered mild cases. (3) Hospitalized participants and patients admitted to the intensive care unit without requiring mechanical ventilation were defined as moderate cases. (4) Participants were considered critical cases if they required mechanical ventilation or extracorporeal membrane oxygenation (ECMO) due to respiratory failure. (5) All patients who died as a result of the infection were classified as deceased cases.

For the purpose of the regression analysis we chose to dichotomize the outcome variable given the small numbers in some of the outcome categories. Consequently, we pooled all cases that were moderate, critical or deceased as hospitalized cases. Vice versa all asymptomatic and mild cases were pooled into the non-hospitalized group for the regression analysis.

**Statistics.**   The analyses were performed using the R statistical language (version 4.0 or higher) on Windows and macOS, and Microsoft Excel 2018 (version 16.16.14 or higher). The statistical analysis plan is available upon request.

We generated a detailed descriptive summary of the study population structure and subgroups with the appropriate measures of central tendency and spread. We compared the demographic structure of our study population to the structure of alle recorded cases in the study area using Pearson's Chi-squared test to assess the representativeness of our study. Unfortunately, recorded cases were only available through the public health authority with information on age, not sex. Our subgroup analyses were constructed around disease severity

which is described in the section above. We performed a post-hoc analysis of age distribution before and after the lock-down in which we decided to set the cut-off value at April 1st, 2020, because the German borders were closed on March 15, 2020. Considering an incubation period of 14 days, the cut-off date of April 1st will likely have excluded all travelers in the second group.

For the inferential statistics, we constructed a multivariable logistic regression model to identify potential risk factors of hospitalization. We identified the potential predictors for the multivariable model through a series of univariate logistic regression models and selected those predictors for the multivariate model that appeared to have a low probability of error ($p < 0.1$) in the univariate models. All univariate predictors with $p < 0.1$ were included in the final multivariate model. As predictors we assessed age as a continuous variable, sex, smoking as a continuous variable using pack years, living with children (age <18), hypertension (yes/no), coronary heart disease (CHD; yes/no), diabetes (type 1 or type 2; yes/no) and lung conditions (yes/no). Lung conditions were defined as a combined variable of either COPD, asthma treated with medications, any other lung disease or previously performed lung surgery. We decided to assess the variable age in a linear relation to allow for easier interpretation and dissemination of the results and because the focus of this paper is primarily of exploratory nature and not predictive.

Secondly, we estimated the influence of the same covariates on the time from symptom onset to hospitalization with a Cox proportional hazard ratio (HR) model. We manually censored all non-hospitalized patients at 14 days after symptom onset, since the majority of patients get hospitalized within 7 days [19].

### Study approval

The institutional ethics board of the University Hospital in Heidelberg approved this study (S-179/2020). Prior to the inclusion in the study, written informed consent was received from each participant. For data protection purposes, all participants were assigned a study ID to ensure pseudonymization.

## Results

By June 30th, 2020, the public health authority in Heidelberg had registered 1,293 SARS-CoV-2 cases in the region with completed quarantine. From these registered cases 166 were hospitalized and a total of 47 patients had died as a result of the infection [20].

Of the registered patients, 142 either refused to give informed consent to the study or were not responding to our inquiry. Thus, our study included 1,151 participants with laboratory-confirmed SARS-CoV-2 infection. Subsequently, 254 participants either withdrew consent or did not submit the questionnaire, leaving a total of 897 patients in the final analysis (69.4% of confirmed cases; Fig 1).

We present clinical and demographic characteristics in Tables 1 and 2. Most patients had from mild symptoms (713, 79.5% of the cohort; 63.3% of all patients not hospitalized among 1,293 cases in the district overall took part in the study). Only a minority of patients remained asymptomatic (54, 6.0%). Altogether, 97 participants (10.8%) were admitted to a hospital without requiring mechanical ventilation (moderate cases), and 16 participants (1.8%) were hospitalized requiring mechanical ventilation (critical cases). In total, 78.3% of all hospitalized cases in the district took part in the study. We observed 17 (1.9%) deaths in the study due to COVID-19 (this made up 36.2% of the total 47 deaths among all 1,293 cases observed in the district). Aside from the age group 70–79 which was slightly underrepresented, (5.8% in our

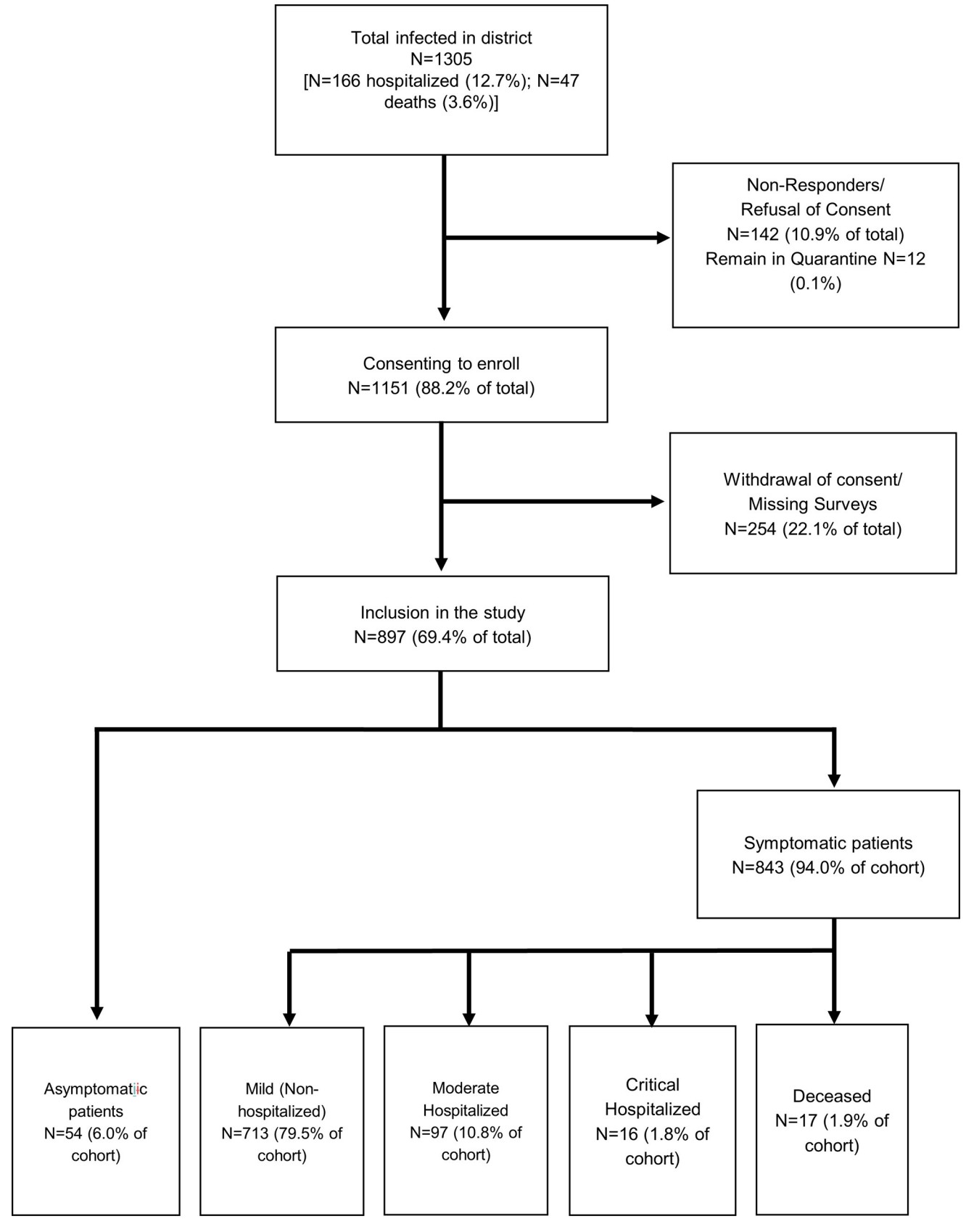

**Fig 1. Study diagram indicating the recruitment process.** Depicting the recruitment process of participants leaving a total of 897 (69.4%) data sets in the final analysis.

study vs. 8.4% among all infected) our study sample represented the age distribution of all SARS-CoV-2 infections in the district at the time of the study (S1 Table).

The mean age of the study population was 47 years (SD 17.5) with a range from 2 to 97. The age average did not change before and after the national lock-down starting on March 15, 2020 [21]. Age increased with rising severity of disease from 52.3 years (SD 20.7) in the mild group to 62.2 years (SD 13.3) in the critical group and 79.1 years (SD 11.8) among the deceased. A substantially higher proportion of participants aged ≥ 70 was admitted to a hospital (28.3%) in comparison to 5.5% of participants below the age of 50. Participants suffering moderate illness were on average 14 years older than participants showing a mild course of disease. Critical cases were on average 18 years older than patients with mild disease.

Around half (453, 50.5%) of the participants were female but the hospitalized patients were predominantly of male sex (58.8% in moderate, 75.0% in critical cases) (Table 1, Fig 2). Smokers or former smokers comprised 36.6% of participants.

**Table 1. Demographics and clinical characteristics.**

| | All patients | Asymptomatic patients | Mild (symptomatic outpatients) | Moderate (hospitalized) | Critical (ventilation) | Deceased |
|---|---|---|---|---|---|---|
| n (%) | 897 (100) | 54 (6.0) | 713 (79.5) | 97 (10.8) | 16 (1.8) | 17 (1.9) |
| Age (years) | | | | | | |
| Mean (SD)–yr | 47.03 (17.5) | 52.3 (20.7) | 44.0 (16.0) | 58.1 (15.9) | 62.2 (13.3) | 79.1 (11.8) |
| Distribution n (%) | | | | | | |
| 0–17 | 17 (1.9) | 0 (0.0) | 17 (2.4) | 0 (0.0) | 0 (0.0) | 0 (0.0) |
| 18–49 | 460 (52.3) | 23 (42.6) | 410 (57.5) | 23 (23.7) | 3 (18.8) | 1 (5.9) |
| 50–59 | 202 (22.5) | 7 (13.0) | 162 (22.7) | 28 (28.9) | 5 (31.3) | 0 (0.0) |
| 60–69 | 112 (12.5) | 9 (16.7) | 78 (10.9) | 21 (21.7) | 3 (18.8) | 1 (5.9) |
| 70–79 | 68 (7.6) | 10 (18.5) | 34 (4.8) | 17 (17.5) | 3 (18.8) | 4 (23.5) |
| >80 | 36 (4.0) | 5 (9.3) | 10 (1.4) | 8 (8.3) | 2 (12.5) | 11 (64.7) |
| Sex n (%) | | | | | | |
| Female | 453 (50.5) | 30 (55.6) | 376 (52.7) | 40 (41.2) | 4 (25.0) | 3 (17.7) |
| Male | 441 (49.2) | 24 (44.4) | 334 (46.8) | 57 (58.8) | 12 (75.0) | 14 (82.3) |
| No answer | 3 (0.3) | 0 (0.0) | 3 (0.4) | 0 (0.0) | 0 (0.0) | 0 (0.0) |
| Participants living with children <18 years n (%) | 269 (30.0) | 11 (20.4) | 236 (33.1) | 18 (18.6) | 4 (25.0) | 0 (0.0) |
| Source of Transmission/ Exposure n (%) | | | | | | |
| 1 Not sure | 215 (24.0) | 18 (33.3) | 155 (21.8) | 31 (32.0) | 7 (43.8) | 4 (23.5) |
| 2 Social contact | 210 (23.4) | 11 (20.4) | 184 (25.8) | 12 (12.4) | 2 (12.5) | 1 (5.9) |
| 3 Work | 180 (20.1) | 8 (14.8) | 154 (21.6) | 18 (18.6) | 0 (0) | 0 (0.0) |
| 4 University, school, kindergarten | 25 (2.8) | 0 (0.0) | 24 (3.4) | 1 (1.0) | 0 (0) | 0 (0.0) |
| 5 Travel | 83 (9.3) | 6 (11.1) | 68 (9.5) | 8 (8.3) | 0 (0) | 1 (5.9) |
| 6 Other | 146 (16.3) | 9 (16.7) | 125 (17.5) | 10 (10.3) | 2 (12.5) | 0 (0) |
| No answer | 38 (4.2) | 2 (3.7) | 3 (0.4) | 17 (17.5) | 5 (31.3) | 11 (64.7) |

Baseline characteristics of 897 participants with coronavirus disease 19 stratified by level of severity.

**Table 2. Clinical course of disease.**

| | All patients | Asymptomatic patients | Mild (symptomatic outpatients) | Moderate (hospitalized) | Critical (ventilation) | Deceased |
|---|---|---|---|---|---|---|
| n (%) | 897 (100) | 54 (6.0) | 713 (79.5) | 97 (10.8) | 16 (1.8) | 17 (1.9) |
| Symptoms[#] | | | | | | |
| n (%) | | | | | | |
| 1. Fever | 481 (53.6) | Asymptomatic | 374 (52.5) | 82 (84.5) | 13 (81.3) | 12 (70.6) |
| 2. Cough | 552 (61.5) | | 457 (64.1) | 72 (74.2) | 9 (56.3) | 14 (82.4) |
| 3. Sputum | 79 (8.8) | | 75 (10.5) | 10 (10.3) | 3 (18.8) | 1 (5.9) |
| 4. Sore throat | 306 (32.1) | | 269 (37.7) | 32 (33.0) | 5 (31.3) | 1 (5.9) |
| 5. Dyspnea | 181 (20.1) | | 113 (15.8) | 50 (51.5) | 10 (62.5) | 8 (47.1) |
| 6. Muscle pain | 279 (31.1) | | 242 (33.9) | 27 (27.8) | 8 (50.0) | 2 (11.8) |
| 7. Limb pain | 432 (48.1) | | 373 (52.3) | 47 (48.5) | 10 (62.5) | 2 (11.8) |
| 8. Fatigue | 586 (65.3) | | 524 (73.5) | 64 (66.0) | 9 (56.3) | 11 (64.7) |
| 9. Headache | 513 (57.2) | | 450 (63.1) | 53 (54.6) | 8 (50.0) | 2 (11.8) |
| 10. Runny nose | 270 (30.1) | | 253 (35.5) | 15 (15.5) | 2 (12.5) | 0 (0.0) |
| 11. Chest pain | 154 (17.1) | | 135 (18.9) | 15 (15.5) | 2 (12.5) | 2 (11.8) |
| 12. Diarrhea | 212 (23.6) | | 173 (24.3) | 29 (29.9) | 5 (31.3) | 5 (29.4) |
| 13. Nausea | 96 (10.7) | | 77 (10.8) | 14 (14.4) | 3 (18.8) | 2 (11.8) |
| 14. Change in taste | 537 (59.8) | | 480 (67.3) | 50 (51.5) | 6 (37.5) | 1 (5.9) |
| 15. Other | 304 (33.9) | | 259 (36.3) | 20 (20.6) | 4 (25.0) | 2 (11.8) |
| Comorbidities n (%) | | | | | | |
| Cardiac disease | 197 (22.0) | 14 (25.9) | 119 (16.7) | 39 (40.2) | 12 (75.0) | 13 (76.5) |
| Hypertension | 152 (17.0) | 9 (16.7) | 98 (13.7) | 29 (29.9) | 6 (37.5) | 10 (58.8) |
| CHD | 19 (2.1) | 2 (3.7) | 7 (1.0) | 2 (2.1) | 2 (12.5) | 6 (35.3) |
| Lung disease | 124 (13.8) | 8 (14.8) | 87 (12.2) | 22 (22.7) | 4 (25.0) | 3 (17.7) |
| Kidney disease | 24 (2.7) | 3 (5.6) | 13 (1.8) | 4 (4.1) | 2 (12.5) | 2 (11.8) |
| Liver disease | 13 (1.5) | 1 (1.9) | 6 (0.8) | 5 (5.2) | 1 (6.3) | 0 (0.0) |
| Diabetes mellitus | 44 (4.9) | 4 (7.4) | 25 (3.5) | 9 (9.3) | 4 (25.0) | 2 (11.8) |
| Autoimmune disorder | 52 (5.8) | 3 (5.6) | 40 (5.6) | 8 (8.3) | 1 (6.3) | 0 (0.0) |
| HIV positive | 0 (0.0) | 0 (0.0) | 0 (0.0) | 0 (0.0) | 0 (0.0) | 0 (0.0) |
| Malignancy | 49 (5.5) | 5 (9.3) | 26 (3.7) | 14 (14.4) | 1 (6.3) | 3 (17.7) |
| Other | 168 (19) | 8 (14.8) | 129 (18.1) | 27 (27.8) | 5 (31.3) | 9 (52.9) |
| Medication n (%) | | | | | | |
| Any | 346 (38.6) | 20 (37.0) | 271 (38.0) | 40 (41.2) | 10 (62.5) | 5 (29.4) |
| NSAR | 71 (7.9) | 7 (13.0) | 51 (7.2) | 8 (8.3) | 2 (12.5) | 3 (17.7) |
| ACEI | 52 (5.8) | 4 (7.4) | 30 (4.2) | 12 (12.4) | 5 (31.3) | 1 (5.9) |
| AT1-Inhibitors | 82 (9.1) | 6 (11.1) | 60 (8.4) | 13 (13.4) | 3 (18.8) | 0 (0.0) |
| Immunosuppressants | 42 (4.7) | 1 (1.9) | 32 (4.5) | 9 (9.3) | 0 (0.0) | 0 (0.0) |
| Chemotherapy | 1 (0.1) | 0 (0.0) | 0 (0.0) | 1 (1.0) | 0 (0.0) | 0 (0.0) |
| Smoking n (%) | | | | | | |
| Never | 533 (59.4) | 40 (74.1) | 442 (62.3) | 43 (44.3) | 5 (31.3) | 3 (17.7) |
| Former | 246 (27.4) | 8 (14.8) | 198 (27.8) | 32 (33.0) | 6 (37.5) | 2 (11.8) |
| Current | 82 (9.1) | 5 (9.3) | 71 (10.0) | 5 (5.2) | 0 (0.0) | 1 (5.9) |
| Missing | 36 (4.0) | 1 (1.9) | 2 (0.3) | 17 (17.5) | 5 (31.3) | 11 (64.7) |
| Days from illness onset to diagnosis Mean (SD) | 6 (4) | x | 6 (4) | 6 (4) | 8 (2) | 7 (4) |

(*Continued*)

**Table 2.** (Continued)

|  | All patients | Asymptomatic patients | Mild (symptomatic outpatients) | Moderate (hospitalized) | Critical (ventilation) | Deceased |
|---|---|---|---|---|---|---|
| Days from illness onset to recovery Mean (SD) | 14 (8) | x | 14 (7) | 19 (9) | 19 (8) | x |

Clinical characteristics and history of 897 participants with coronavirus disease 19 stratified by level of severity. Data are either numbers (percentages) or means (standard deviations) as indicated in the table. Given symptoms are pooled symptoms comprising of initial and developed symptoms. See S2 Table for further details.

The most commonly reported symptoms were fatigue (65.3%), followed by cough (61.5%) and dysgeusia (59.8%) (Table 2; S2 Table). Dyspnea was substantially more frequent in moderate and critical cases than mild ones (51.5%, 62.5% versus 15.8%).

Nearly half (46.6%) of the study population reported at least one underlying comorbidity, the most common was hypertension (152, 17%) and 38.6% (346 patients) of the study population was on regular medication. The presence of comorbidities including heart disease, lung disease and diabetes was more common among patients with a moderate/ critical course of disease versus a mild course (mild 16.7% versus moderate/critical 40.3% / 75.0%; 12.2% versus 22.7% / 25.0%; 3.5% versus 9.3% / 25.0%, respectively; Table 1).

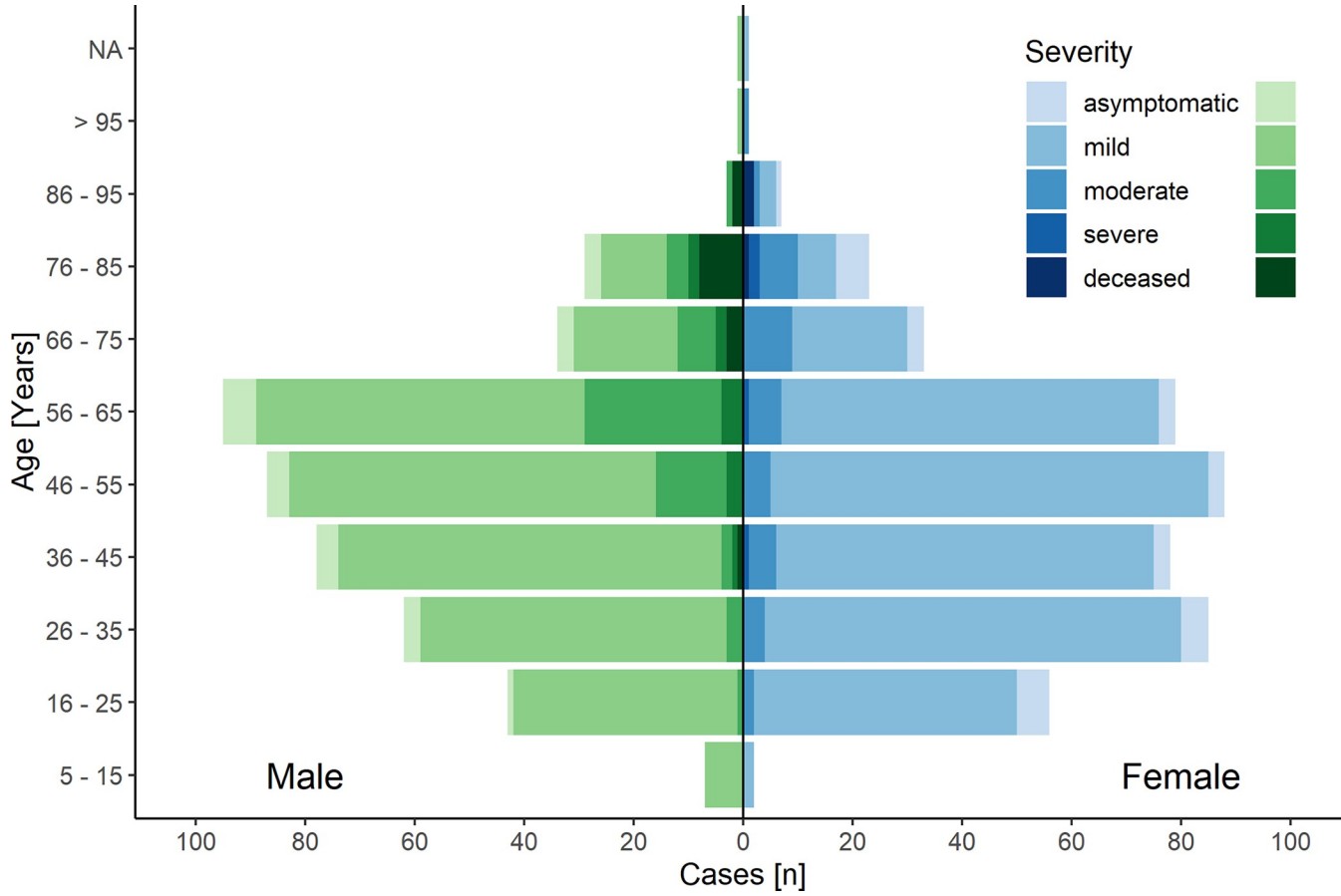

**Fig 2. Severity and age distribution.** Patients with COVID-19 stratified by age and sex. Female n = 453; Male n = 441 The left-hand side (green) shows male and the right-hand side (blue) female participants. The shades of color represent different level of disease severity. The severity of disease increases with darker shades of color. Further details on the group´s composition can be found in Tables 1 and 2.

**Table 3. Risk factors associated with hospital admission.**

| Variables | Non-hospitalized (n = 767) | Hospitalized and deceased (n = 130) | OR | 95% Cis |
|---|---|---|---|---|
| Age Mean (SD) | 44.6 (16.3) | 61.4 (15.0) | 1.05 | 1.04–1.07 |
| Sex n (%) | | | | |
| Female | 406 (52.9) | 47 (36.2) | 0.51 | 0.33–0.77 |
| Lung disease incl. n (%) | | | | |
| • Any lung disease | 95 (12.4) | 29 (22.3) | 3.09 | 1.62–5.88 |
| • Chronic Asthma with medication | | | | |
| • Any lung surgery | | | | |
| Smoking history n (%) | 282 (36.8) | 46 (35.4) | 1.00 | 0.98–1.01 |
| Coronary heart disease n (%) | 9 (1.2) | 10 (7.7) | 1.44 | 0.51–4.07 |
| Hypertension n (%) | 107 (14.0) | 45 (34.6) | 0.98 | 0.59–1.65 |
| Diabetes mellitus Type 1 or 2 n (%) | 29 (3.8) | 15 (11.5) | 1.38 | 0.65–2.91 |
| Living with children n (%) | 247 (32.2) | 22 (16.9) | 0.80 | 0.46–1.37 |

Multivariable logistic regression was conducted to display the effect of potential risk factors (age, sex, comorbidities, smoking history and children living in household) for hospitalization. All moderate, severe and deceased cases were pooled as hospitalized in the context of the regression analysis. All asymptomatic and mild cases were pooled for the outpatient group. n = 897.

The mean duration of disease from symptom onset to resolution of symptoms was 14 days (SD of 8 days) (Table 2). Among the hospitalized population, the mean time from symptom onset to hospitalization was 8 (SD 4), 8 (SD 3) and 7 (SD 4) days for moderate, critical and deceased participants, respectively. The mean length of hospital stay was 10 days (SD 9) in moderate and 24 days (SD 20) in critical cases.

The multivariable logistic regression model identified potential risk factors for disease severity that led to hospitalization (Table 3). The strongest predictors for hospitalization included greater age with every year increase conferring a higher risk (OR 1.05; 95% CI 1.04–1.07) and the presence of lung disease (OR 3.09; 95% CI 1.6–5.9). Female sex (OR 0.51; 95% CI 0.3–0.8), on the other hand, was identified as a protective factor for disease progression. The presence of coronary heart disease and diabetes showed a trend towards being associated with an increased risk of hospitalization with an increased OR of 1.44 (95% CI 0.51–4.07) and OR 1.38 (95% CI 0.65–2.91) of hospitalization, but confidence interval crossed 1. Hypertension was not associated with hospitalization (OR 0.98, 95% CI 0.59–1.65). Living with children (< 18 years of age) in a household demonstrated a trend towards a protective effect (OR 0.80; 95% CI 0.46–1.37) (Table 3).

The multivariable cox regression was built around the same covariates as the logistic regression model to estimate their influence on the time from symptom onset to hospitalization. This cox regression, however, yielded similar results to the logistic regression and considering the limited time frame of the study, offered no added value to our analysis (S3 Table).

## Discussion

This retrospective cohort study provides a comprehensive picture of clinical features and factors associated with severe disease in a primarily symptomatic population infected with SARS-CoV-2 in an administrative district in Southern Germany. The study shows 6% asymptomatic cases, a low percentage of hospitalization (12.9%) and confirms older age, lung conditions and male sex to be associated with greater disease severity.

The study provides a representative picture of COVID-19 in symptomatic patients in the general population in Germany in a district that was moderately affected by the disease

(cumulative prevalence 0.18%) [16, 17]. However, we have to acknowledge that we very likely under-sampled asymptomatic patients, given the difficulty in detecting them despite extensive case finding efforts. Furthermore, we included less than half of the cases with fatal outcomes due to difficulties with gathering information from their close relatives in times of grief and severe stress. We therefore decided to use hospitalization with or without intubation and subsequent death as our main outcome in the analysis.

Previous studies primarily described characteristics of hospitalized patients [6, 11, 13, 14, 22–31] leading to limited data regarding mildly affected or asymptomatic patients. While the percentages of mild cases in our study were similar to the results of Wu and McGoogan, who described 72,314 cases (80% mild versus 79% in our study) [10], asymptomatic cases were reported in only 1% of their findings and severe and critical cases in 19% (as opposed to 6% and 4% in our study, respectively). This may be attributed to the fact that the study population of Wu and McGoogan mostly included patients from Hubei province and the rapid spread led to exhausted health care resources causing asymptomatic and less severely affected patients to be underrepresented in their report.

While the number of asymptomatic cases in our study is larger compared to the Wu and McGoogan study, it is likely that the actual percentage in the population is even higher based on other studies of complete cohorts and also modelling exercises [6, 32]. Also, while the number of hospitalizations was lower in our study compared to Wu and McGoogan and others, we postulate that Germany´s well-established health care infrastructure combined with the uncertainty about the novel disease may have allowed for an increased hospitalization and ICU admission out of precaution beyond of what would be considered necessary with a known disease.

The case fatality ratio was 3.6% in the Rhein-Neckar-Region [33]. This case fatality ratio was, however, lower than the overall death rate in Germany reported at 4.6% [34]. While differences are small, the discrepancy could be due to the patient populations in Southern Germany versus the whole of Germany as in Southern Germany more young people who returned from skiing in endemic regions were infected early in the epidemic. However, in our analysis of age before and after the lock down, we found no substantial differences to support this theory (S1 Fig).

Consistent with other studies, we identified fatigue (65.3%) and cough (61.5%) as the most predominant symptoms in patients suffering from COVID-19 [27, 31, 35]. Additionally, we found headache (57.2%) and dysgeusia (59.8%) among the most common symptoms. However, no symptom or symptom constellation appears to be frequent and specific enough to consider it for screening. Further studies considering advanced analyses of large datasets with symptoms of COVID-19 patients and randomly sampled cohorts of patients without COVID-19 at different times of the year should be considered to develop screening algorithms.

Identifying risk factors of patients prone to develop severe disease would help to focus medical surveillance and apply treatment at an early stage [13]. We identified older age, male sex as well as lung conditions as potential risk factors for a severe course of infection in our logistic regression. Our findings are consistent with previous studies who have described age, male sex, and lung disease as risk factors [14, 23, 27]. However, in contrast to other studies, we did not identify hypertension and diabetes mellitus as significant independent risk factor of a severe course of COVID-19, although odds ratios were increased. An increased risk would be supported by the pathophysiology of the virus. SARS-CoV-2 uses ACE2 receptors as cellular key entries. The virus then causes a down regulation of the same receptor leading to an increased permeability of the pulmonal vascular system [36]. As a result, pulmonary injuries may exacerbate and induce a more severe course of disease in patients suffering from hypertension or diabetes mellitus.

Unfortunately, the we were unable to collect reliable data on weight in our study. However, another study importantly demonstrated the relevance of obesity towards an increased risk of severe disease, hospital admission and mortality which we could not account for in our study [37].

Interestingly enough, the presence on children in a household showed and odds ratio below 1. Even though our correlation was not statistically significant, a recent publication observed similar findings. In a cohort study from Scotland among 300,000 adults living in a household, the risk of testing positive for SARS-CoV-2 was slightly lower for individuals living with young children after adjusting for potential confounders [38]. However, large data sets from other countries need to be analyzed to address this with more sincerity as these findings could be of importance for informing policy on school openings.

While our study is able to provide a comprehensive overview of the disease and the predictors of severe disease among Sars-CoV-2 infected patients, it also has several limitations. First, due to the retrospective study design, some patients filled out the questionnaire several weeks after being tested positive or showing symptoms which may lead to a recall bias. Second, patients not speaking German were not able to be included in the study due to the informed consent being available only in German. Third, relatives of deceased patients and nursing home residents were harder to reach and less open to participate in the study which leads to an underrepresentation of these groups. Fourth, patients suffering severe courses of disease were hospitalized for an extended period of time and rehabilitation measures were applied afterwards, making these patients difficult to reach. This may have led to a selection bias. Besides disease severity, hospitalization is also influenced by factors such as compliance of patients, hospital management or affordability of the health care system tailored to the respective setting which may have affected event occurrence as well. Fifth, 30% of patients in the registry could not be reached, withdrew consent or did not return the survey leading to missing data.

## Conclusion

This analysis of representative COVID-19 cases confirms age, male sex and preexisting lung conditions but not cardiovascular disease as risk factors for severe illness.

Health care systems should prepare for about 15% of patients to suffer moderate or critical illness.

## Supporting information

**S1 Table. Data comparison of study population vs. overall infected population.** To assess the representativeness of our study population in comparison with the overall population infected with SARS-CoV-2, we compared the age groups sorted by age groups using the Chi-Square test.
(PDF)

**S2 Table. Initial and subsequent symptoms.** Participants were asked to specify symptoms they suffered in the initial phase of the infection and symptoms which developed during the course of disease. n = 897.
(PDF)

**S3 Table. Cox regression, influence of variables on time to hospitalization.** Multivariable Cox proportional hazard model analyzing age, sex, comorbidities and smoking history (in pack years). n = 897.
(PDF)

**S1 Fig. Age distribution among study population.** In the early part of the epidemic many cases returned from skiing holidays. We display the age distribution before and after the border closure considering 14 days of an incubation period (i.e. April 1st, 2020). n = 897. (PNG)

**S1 File. Questionnaire.** Copy of the questionnaire distributed to consenting participants. (PDF)

**S1 Text. Additional methods.** Description of variables. (PDF)

## Acknowledgments

We thank all the participants of the study and the facilitators in the retirement homes for their commitment and contribution to the study. This study would not have been possible without them. And we are thankful to colleagues in the Public Health Authority Rhein-Neckar, especially, Anne Kühn and Nadja Knis, for their efforts to provide the needed support. Additionally, we want to thank Shannon McMahon for providing input into the questionnaire design.

## Author Contributions

**Conceptualization:** Yannis Herrmann, Aurélia Souares, Claudia M. Denkinger.

**Data curation:** Yannis Herrmann, Niall Brindl, Philip J. Kitchen, Lukas Rädeker.

**Formal analysis:** Yannis Herrmann, Tim Starck, Lisa Köppel, Claudia M. Denkinger.

**Investigation:** Yannis Herrmann, Niall Brindl, Philip J. Kitchen, Lukas Rädeker, Jakob Sebastian, André L. Mihaljevic, Uta Merle, Theresa Hippchen, Felix Herth.

**Methodology:** Yannis Herrmann, Tim Starck, Lisa Köppel, Claudia M. Denkinger.

**Project administration:** Yannis Herrmann, Lukas Rädeker, Frank Tobian, Claudia M. Denkinger.

**Resources:** Aurélia Souares.

**Software:** Frank Tobian.

**Supervision:** Lisa Köppel, Claudia M. Denkinger.

**Validation:** Yannis Herrmann, Tim Starck, Niall Brindl, Philip J. Kitchen, Lukas Rädeker, Jakob Sebastian, Lisa Köppel, Britta Knorr, Andreas Welker, Claudia M. Denkinger.

**Visualization:** Tim Starck.

**Writing – original draft:** Yannis Herrmann, Tim Starck.

**Writing – review & editing:** Yannis Herrmann, Tim Starck, Niall Brindl, Philip J. Kitchen, Lukas Rädeker, Jakob Sebastian, Lisa Köppel, Frank Tobian, Aurélia Souares, André L. Mihaljevic, Uta Merle, Theresa Hippchen, Felix Herth, Britta Knorr, Andreas Welker, Claudia M. Denkinger.

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
