## [Decision Letter · Decision Letter 0]

10 May 2021

PONE-D-21-06695

Description and analysis of representative COVID-19 cases – a retrospective cohort study

PLOS ONE

Dear Dr. Denkinger,

Thank you for submitting your manuscript to PLOS ONE. After careful consideration, we feel that it has merit but does not fully meet PLOS ONE’s publication criteria as it currently stands. Therefore, we invite you to submit a revised version of the manuscript that addresses the points raised during the review process.

We look forward to receiving your revised manuscript.

Kind regards,

Tai-Heng Chen, M.D.

Academic Editor

PLOS ONE

[This work was supportedby internal COVID funds of the Heidelberg University Hospital]

 [The authors received no specific funding for this work.]

Reviewers' comments:

Reviewer's Responses to Questions

**Comments to the Author**

1. Is the manuscript technically sound, and do the data support the conclusions?

Reviewer #2: Yes

Reviewer #3: Partly

Reviewer #4: Yes

2. Has the statistical analysis been performed appropriately and rigorously? 

Reviewer #2: Yes

Reviewer #3: No

Reviewer #4: Yes

3. Have the authors made all data underlying the findings in their manuscript fully available?

Reviewer #2: Yes

Reviewer #3: Yes

Reviewer #4: Yes

4. Is the manuscript presented in an intelligible fashion and written in standard English?

Reviewer #2: Yes

Reviewer #3: Yes

Reviewer #4: Yes

5. Review Comments to the Author

Reviewer #2: The authors conducted a retrospective survey on the COVID-19 patients in Southern German, described clinical features and assessed potential risk factors for hospitalization. Overally, the article is well written and well designed. In my opinion, it need only some minor revisions.

1.Statistics: what kind of multivariate logistic regression model was performed, backward stepwise or others？

2.I thinck hospitalization is influenced potentially by many factors, e.g., disease severity, compliance of patients, the affordability of health care system, and even management policy in each country. For example, in China, even asymptomatic and mild COVID-19 patients are hospitalized. So, I suggest this should be discussed in section of limitation.

3.P value is not a trend. So the sentences should be expressed in a more appropriate way (Page 20, line 358, line 364).

Reviewer #3: 1. This is an interesting study but the findings are not of sufficient originality and there is a lack of novelty as compared to a number of similar studies already published on the topic. In addition, there are some concerns need to be thought carefully.

2. Patients enrolled from February 7 to June 30 were included in the follow-up, and the paper needs to explain when the retrospective investigation was conducted.

3. There are major problems with the study design. The study design of this paper is not a retrospective cohort study, not based on the exposure of certain risk factors, but divided the outcomes of the cases into different study levels and collected relevant data retrospectively, which should be a cross-section investigation design and then do case-control study and risk factors analysis.

4. In the study, 30% cases lost to follow-up may affect the distribution of the pathogen of the disease, making the results less reliable. So, suggest to analysis the basic situation of the lost.

5. Regarding to the risk factor analysis, it is more appropriate for the authors to conduct univariate analysis firstly to find meaningful variate and then do multiple logistic regression.

6. The authors should notice that all COVID-19 including are required to hospital for isolation in China.

7. Line 212-213, page 19, please check the number of percent and its corresponding type of patients. It is inconsistent with the figure presented in Table 2.

Reviewer #4: General Comments

Thank you very much for choosing me as a reviewer of this manuscript. I think this article is well written and it looks into an important and concerning problem nowadays as is COVID-19. This study is a demographic and epidemiological analysis of a German cohort representative of an administrative district. They found that age, male sex and pre-existing lung conditions were associated with severe illness. The statistical analysis is well done and well discussed.

Although is difficult to assess the real incidence of SARS-CoV-2 infection in this cohort and asymptomatic patients’s rate could be infra estimated, they analyzed 69% of all COVID-19 registered cases with an important underrepresentation of the group of severe disease. All these limitations are reported at the discussion by the authors.

Finally, I have provided some specific comments point by point.

Discussion

Data regarding BMI index is missing in the study and it has also been related to worse outcomes in other large community-based cohort study (Gao M, Piernas C, Astbury NM, Hippisley-Cox J, O'Rahilly S, Aveyard P, Jebb SA. Associations between body-mass index and COVID-19 severity in 6•9 million people in England: a prospective, community-based, cohort study. Lancet Diabetes Endocrinol. 2021 Apr 28:S2213-8587(21)00089-9). Please, add this comment to the discussion section.

Supplementary material

Please translate the questionnaire into English

6. PLOS authors have the option to publish the peer review history of their article (what does this mean?). If published, this will include your full peer review and any attached files.

Reviewer #2: **Yes: **Liang Chen

Reviewer #3: No

Reviewer #4: No

---

## [Author Response · Author response to Decision Letter 0]

22 May 2021

Journal comments to the author:

Response: thank you for this comment. We have made appropriate adaptions throughout the document. 

[This work was supportedby internal COVID funds of the Heidelberg University Hospital]

 [The authors received no specific funding for this work.]

Response: we excluded the funding statement in the manuscript and wish to keep the following statement 

(“This work was supported by internal COVID funds of the Heidelberg University Hospital.”)

Response: thank you for this remark. Appropriate changes have been incorporated throughout. 

Reviewer 1:

1. The authors conducted a retrospective survey on the COVID-19 patients in Southern German, described clinical features and assessed potential risk factors for hospitalization. Overally, the article is well written and well designed. In my opinion, it need only some minor revisions.

Response: we appreciate the reviewer’s comment. 

2. Statistics: what kind of multivariate logistic regression model was performed, backward stepwise or others？

Response: thank you for this comment! We did not use any stepwise model selection algorithms but selected the variables for the multivariate analysis from a series of univariate screening models.

We clarified this in lines 134-138

3. I think hospitalization is influenced potentially by many factors, e.g., disease severity, compliance of patients, the affordability of health care system, and even management policy in each country. For example, in China, even asymptomatic and mild COVID-19 patients are hospitalized. So, I suggest this should be discussed in section of limitation.

Response: we appreciate the reviewer’s comment and have added this to the limitation section (line 443-446)

4. P value is not a trend. So the sentences should be expressed in a more appropriate way (Page 20, line 358, line 364)

Response: thank you for this comment, we have changed the sentences (line 417-423)

Reviewer 3:

1. Patients enrolled from February 7 to June 30 were included in the follow-up, and the paper needs to explain when the retrospective investigation was conducted.

Response: thank you for your remark. We added information on the period of conduct (line 60-61). Data collection started on March 19, 2020 and concluded on June 30, 2020.

2. There are major problems with the study design. The study design of this paper is not a retrospective cohort study, not based on the exposure of certain risk factors, but divided the outcomes of the cases into different study levels and collected relevant data retrospectively, which should be a cross-section investigation design and then do case-control study and risk factors analysis.

Response: we appreciate the reviewer’s valuable feedback. A case control study or prospective cohort study might have been of higher value. However, given the dynamics of the early pandemic this was not feasible. 

We believe that or study fills all the formal criteria for a retrospective cohort study in that it:

- Describes a cohort selected by the occurrence of an outcome

- Retrospectively evaluates risk factors that might have contributed to said diseases, i.e. accumulated significantly in our population

Retrospectively assesses said risk factors contribution to the severity of the disease (thus stratifying the populations into groups, e.g. age, smoking status and using this stratification as case-control setup)

3. In the study, 30% cases lost to follow-up may affect the distribution of the pathogen of the disease, making the results less reliable. So, suggest to analysis the basic situation of the lost.

Response: thank you for this comment. We highlighted this limitation of the study at the end of our discussion. (line 454-455) In addition, we compared the age structure of our study population to the overall population in the district tested positive for SARS-CoV-2 to identify potential underrepresentation of age groups. (line 129-131; line 194-197)

4. Regarding to the risk factor analysis, it is more appropriate for the authors to conduct univariate analysis firstly to find meaningful variate and then do multiple logistic regression.

Response: we appreciate the reviewer’s comment and have revised the appropriate methods section to indicate that we had done a univariate analysis first. The section should be clearer now. (Line 139-143)

5. The authors should notice that all COVID-19 including are required to hospital for isolation in China.

Response: we are thankful for this information and have acknowledged this. This is not the case in Germany. 

6. Line 212-213, page 19, please check the number of percent and its corresponding type of patients. It is inconsistent with the figure presented in Table 2

Response: we appreciate that the reviewer pointed this out. We have identified the mistake and included appropriate changes but could not find inconsistencies on page 19.

Reviewer 4:

1. Discussion

Data regarding BMI index is missing in the study and it has also been related to worse outcomes in other large community-based cohort study (Gao M, Piernas C, Astbury NM, Hippisley-Cox J, O'Rahilly S, Aveyard P, Jebb SA. Associations between body-mass index and COVID-19 severity in 6•9 million people in England: a prospective, community-based, cohort study. Lancet Diabetes Endocrinol. 2021 Apr 28:S2213-8587(21)00089-9). Please, add this comment to the discussion section.

Response: thank you for this helpful comment! Unfortunately, the data collected on weight in our study was not reliable, which led us to exclude the information. However, we added this risk factor to our discussion (line 417-424).

2. Supplementary material

Please translate the questionnaire into English

Response: we appreciate the reviewer’s feedback and have added supplement tables that depict the assessed variables of the questionnaire.

---

## [Decision Letter · Decision Letter 1]

16 Jun 2021

PONE-D-21-06695R1

Description and analysis of representative COVID-19 cases – a retrospective cohort study

PLOS ONE

Dear Dr. Denkinger,

Thank you for submitting your manuscript to PLOS ONE. After careful consideration, we feel that it has merit but does not fully meet PLOS ONE’s publication criteria as it currently stands. Therefore, we invite you to submit a revised version of the manuscript that addresses the points raised during the review process.

We look forward to receiving your revised manuscript.

Kind regards,

Tai-Heng Chen, M.D.

Academic Editor

PLOS ONE

Journal Requirements:

Reviewers' comments:

Reviewer's Responses to Questions

**Comments to the Author**

1. If the authors have adequately addressed your comments raised in a previous round of review and you feel that this manuscript is now acceptable for publication, you may indicate that here to bypass the “Comments to the Author” section, enter your conflict of interest statement in the “Confidential to Editor” section, and submit your "Accept" recommendation.

Reviewer #3: (No Response)

2. Is the manuscript technically sound, and do the data support the conclusions?

Reviewer #3: Yes

3. Has the statistical analysis been performed appropriately and rigorously? 

Reviewer #3: Yes

4. Have the authors made all data underlying the findings in their manuscript fully available?

Reviewer #3: Yes

5. Is the manuscript presented in an intelligible fashion and written in standard English?

Reviewer #3: Yes

6. Review Comments to the Author

Reviewer #3: In my opinion, cohort study means classification is made according to exposure factor, generally one factor. For example, if we want to study the association between smoking (exposure factor) and lung cancer (outcome variable), the study population are classified as smoking group and non-smoking group according to their smoking history status. And then follow them, observe whether they develop lung cancer or not after a period of times. Usually, only one exposure factor is used as the classified variable. Conversely, the outcome variables may be more than one. In the above mentioned example, aside from lung cancer, the outcome variables may be hypertention, stroke or others. Here, we calculate the relative ration (RR) rather than the odds ratio (OR) to show the degree of association between the two factors (here is smoking and lung cancer). So, if the author declared that they used a cohort study, please explain what the exposure variable and outcome variable were in their study.

7. PLOS authors have the option to publish the peer review history of their article (what does this mean?). If published, this will include your full peer review and any attached files.

Reviewer #3: No

---

## [Author Response · Author response to Decision Letter 1]

5 Jul 2021

Journal Comments to the Author 

Response: 

We have checked our references and ensured only appropriate references were included. We did not include retracted publications. We revised two preprint references because both studies had undergone peer-review and were published in journals. The first reference titled “A demographic scaling model for estimating the total number of COVID-19 infections” by Bohk-Ewald et al. was published in the International Journal of Epidemiology. The second one named “Prevalence of SARS-CoV-2 Infection in Residents of a Large Homeless Shelter in Boston” by Baggett et al was published in JAMA. 

Reviewer 3 Comments to the Author 

1. In my opinion, cohort study means classification is made according to exposure factor, generally one factor. For example, if we want to study the association between smoking (exposure factor) and lung cancer (outcome variable), the study population are classified as smoking group and non-smoking group according to their smoking history status. And then follow them, observe whether they develop lung cancer or not after a period of times. Usually, only one exposure factor is used as the classified variable. Conversely, the outcome variables may be more than one. In the above mentioned example, aside from lung cancer, the outcome variables may be hypertention, stroke or others. Here, we calculate the relative ration (RR) rather than the odds ratio (OR) to show the degree of association between the two factors (here is smoking and lung cancer). So, if the author declared that they used a cohort study, please explain what the exposure variable and outcome variable were in their study.

Response: 

We appreciate the reviewer’s feedback and generally agree with the raised points and assessment. Coherent with the design of a cross-sectional study, our study was constructed to show correlations between an exposure and outcome variable. Due to the retrospective design, data for both were collected at the same time and the Odds ratio was calculated based on a logistic regression model. Nonetheless, we still are of the opinion that this study could also be described as a cohort study due to the reasons mentioned below.

In their textbook on medical statistics, Aviva and Sabin define a (retrospective) cohort study as follows: “A cohort study takes a group of individuals and usually follows them forward in time, the aim being to study whether exposure to a particular aetiological factor will affect the incidence of a disease outcome in the future. If so, the factor is generally known as a risk factor for the disease outcome. […] Although most cohort studies are prospective, historical cohorts are occasionally used: these are identified retrospectively and relevant information relating to outcomes and exposures of interest up to the present day ascertained using medical records and memory.” 

(Petrie, Aviva, and Caroline Sabin. Medical Statistics at a Glance, John Wiley & Sons, Incorporated, 2009. ProQuest Ebook Central, http://ebookcentral.proquest.com/lib/ub-heidelberg/detail.action?docID=1561073.

Created from ub-heidelberg on 2021-06-24 15:30:09.)

We believe that this definition of a historical or retrospective cohort applies to our study because a defined group (in this case patients who tested positive for SARS-CoV-2) was followed backwards in time to identify whether the exposure of an aetiological factor (i.e. age, sex heart disease, lung disease, diabetes, smoking history, living with children) had an effect on the defined outcome (i.e. hospitalization, severity of disease).

Instead of classifying by exposure variables (i.e. age, sex heart disease, lung disease, diabetes, smoking history, living with children) as usually done in cohort studies, we decided to classify the population by the outcome variable (hospitalization and no hospitalization) due to the relatively large number of variables considered in the analysis. This in our opinion allowed a better and more structured way of presenting our results to the reader without defying the essence of a cohort study.

Because of the complex demographic of our study population, we needed to adjust for confounding factors. To do so, we created a logistic regression model for our (binary) exposure variables and therefore, present the results as odds ratios. 

Even though we, alongside other reviewers, believed to have fulfilled the criteria for a retrospective cohort study, we acknowledge that our study also meets criteria for a cross-sectional study design. If deemed necessary by the editor, we will change the title and methods to cross-sectional instead of cohort.

---

## [Decision Letter · Decision Letter 2]

19 Jul 2021

Description and analysis of representative COVID-19 cases – a retrospective cohort study

PONE-D-21-06695R2

Dear Dr. Denkinger,

We’re pleased to inform you that your manuscript has been judged scientifically suitable for publication and will be formally accepted for publication once it meets all outstanding technical requirements.

Kind regards,

Tai-Heng Chen, M.D.

Academic Editor

PLOS ONE

Reviewers' comments:

Reviewer's Responses to Questions

**Comments to the Author**

1. If the authors have adequately addressed your comments raised in a previous round of review and you feel that this manuscript is now acceptable for publication, you may indicate that here to bypass the “Comments to the Author” section, enter your conflict of interest statement in the “Confidential to Editor” section, and submit your "Accept" recommendation.

Reviewer #3: (No Response)

2. Is the manuscript technically sound, and do the data support the conclusions?

Reviewer #3: (No Response)

3. Has the statistical analysis been performed appropriately and rigorously? 

Reviewer #3: (No Response)

4. Have the authors made all data underlying the findings in their manuscript fully available?

Reviewer #3: (No Response)

5. Is the manuscript presented in an intelligible fashion and written in standard English?

Reviewer #3: (No Response)

6. Review Comments to the Author

Reviewer #3: (No Response)

7. PLOS authors have the option to publish the peer review history of their article (what does this mean?). If published, this will include your full peer review and any attached files.

Reviewer #3: No

---

## [Editor Report · Acceptance letter]

23 Jul 2021

PONE-D-21-06695R2 

Description and analysis of representative COVID-19 cases – a retrospective cohort study 

Dear Dr. Denkinger:

I'm pleased to inform you that your manuscript has been deemed suitable for publication in PLOS ONE. Congratulations! Your manuscript is now with our production department. 

Kind regards, 

on behalf of

Dr. Tai-Heng Chen 

Academic Editor

PLOS ONE